# Reduced Cross-Sectional Muscle Growth Six Months after Botulinum Toxin Type-A Injection in Children with Spastic Cerebral Palsy

**DOI:** 10.3390/toxins14020139

**Published:** 2022-02-14

**Authors:** Nathalie De Beukelaer, Guido Weide, Ester Huyghe, Ines Vandekerckhove, Britta Hanssen, Nicky Peeters, Julie Uytterhoeven, Jorieke Deschrevel, Karen Maes, Marlies Corvelyn, Domiziana Costamagna, Ghislaine Gayan-Ramirez, Anja Van Campenhout, Kaat Desloovere

**Affiliations:** 1Neurorehabilitation Group, Department of Rehabilitation Sciences, KU Leuven, 3000 Leuven, Belgium; weideguido@gmail.com (G.W.); ester.huyghe@kuleuven.be (E.H.); ines.vandekerckhove@kuleuven.be (I.V.); britta.hanssen@kuleuven.be (B.H.); nicky.peeters@kuleuven.be (N.P.); julie.uytterhoeven@kuleuven.be (J.U.); domiziana.costamagna@kuleuven.be (D.C.); kaat.desloovere@uzleuven.be (K.D.); 2Laboratory for Myology, Faculty of Behavioural and Movement Sciences, Vrije Universiteit Amsterdam, Amsterdam Movement Sciences, 1081 Amsterdam, The Netherlands; 3Department of Rehabilitation Science, Ghent University, 9000 Ghent, Belgium; 4Laboratory of Respiratory Diseases and Thoracic Surgery, Department of Chronic Diseases and Metabolism, KU Leuven, 3000 Leuven, Belgium; jorieke.deschrevel@kuleuven.be (J.D.); karen.maes@kuleuven.be (K.M.); ghislaine.gayan-ramirez@kuleuven.be (G.G.-R.); 5Stem Cell Biology and Embryology, Department of Development and Regeneration, KU Leuven, 3000 Leuven, Belgium; marlies.corvelyn@kuleuven.be; 6Unit of Pediatric Orthopedics, Department of Orthopedics, University Hospitals Leuven, 3000 Leuven, Belgium; anja.vancampenhout@uzleuven.be; 7Department of Development and Regeneration, KU Leuven, 3000 Leuven, Belgium; 8Clinical Motion Analysis Laboratory, University Hospitals Leuven, Pellenberg, 3212 Leuven, Belgium

**Keywords:** botulinum neurotoxin type-A, spastic cerebral palsy, muscle growth rate, 3D freehand ultrasound

## Abstract

Botulinum Neurotoxin type-A (BoNT-A) injections are widely used as first-line spasticity treatment in spastic cerebral palsy (SCP). Despite improved clinical outcomes, concerns regarding harmful effects on muscle morphology have been raised. Yet, the risk of initiating BoNT-A to reduce muscle growth remains unclear. This study investigated medial gastrocnemius (MG) morphological muscle growth in children with SCP (*n* = 26, median age of 5.2 years (3.5)), assessed by 3D-freehand ultrasound prior to and six months post-BoNT-A injections. Post-BoNT-A MG muscle growth of BoNT-A naive children (*n* = 11) was compared to (a) muscle growth of children who remained BoNT-A naive after six months (*n* = 11) and (b) post-BoNT-A follow-up data of children with a history of BoNT-A treatment (*n* = 15). Six months after initiating BoNT-A injection, 17% decrease in mid-belly cross-sectional area normalized to skeletal growth and 5% increase in echo-intensity were illustrated. These muscle outcomes were only significantly altered when compared with children who remained BoNT-A naive (+4% and −3%, respectively, *p* < 0.01). Muscle length growth persevered over time. This study showed reduced cross-sectional growth post-BoNT-A treatment suggesting that re-injections should be postponed at least beyond six months. Future research should extend follow-up periods investigating muscle recovery in the long-term and should include microscopic analysis.

## 1. Introduction

Starting from an early age, the medial gastrocnemius (MG) muscle morphology is impaired in children with spastic cerebral palsy (SCP) [1,2]. Indeed, cross-sectional observations using three-dimensional (3D) macroscopic muscle imaging demonstrated MG deficits in muscle volume (MV), anatomical cross-sectional area (aCSA) and muscle lengths in comparison to age-matched typically developing (TD) children [3,4]. Furthermore, primary symptoms of the upper motor neuron lesion, such as spasticity and muscle weakness, predominate the clinical presentation of altered motor functioning (e.g., gait abnormalities) [5,6]. The injection of botulinum neurotoxin type-A (BoNT-A) is widely used within the conventional management of spasticity. This treatment is already prescribed from the ages of two to four years, aiming to reduce focal muscle hyper-excitability [7]. Beneficial outcomes such as reduced muscle spasticity, increased ankle joint range of motion and improved gait have been described when BoNT-A injections are combined with concurrent therapies (e.g., serial casting, ankle–foot orthoses and physiotherapy) [8,9,10,11].

However, concerns regarding harmful effects of BoNT-A exposure on muscle morphology have been raised primarily based on muscle studies in animals and healthy humans [12,13,14,15,16,17]. More specifically, the repeated administration of BoNT-A injections has caused muscle atrophy in rabbits for which the first exposure resulted in the largest reduction [16]. In clinical follow-up studies among ambulatory children with SCP with no previous history of BoNT-A injection (i.e., BoNT-A naive), the neuro-paralytic working mechanism resulted in MG muscle atrophy (defined by reduced MV) of 5.9% and 9.4%, after five (i.e., acute effect) and 13 weeks (i.e., short-term effect), respectively. Furthermore, a six month follow-up study (i.e., medium-term effect) of BoNT-A naive children reported a decrease of 6.8% in MG muscle volume, suggesting hampering of muscle growth as a result of the BoNT-A injection [18].

Since skeletal growth is found to be altered in the SCP population compared to TD children, normalization of muscle outcomes to skeletal growth should be accounted for when investigating potential hampered muscle growth [19,20]. Only one post-BoNT-A follow-up study described the amount of MG atrophy by expressing the change in MV relative to limb growth over the study period. Compared to the baseline normalized MV (nMV), reductions were found at four (−6.7%), 13 (−10.7%) and 25 (−9.8%) weeks post-initiation of BoNT-A injection [18]. Additionally, Barber et al. investigated the muscle growth rate calculated as the ratio of muscle volume per time both within a cross-sectional (i.e., ratio per baseline age) and a longitudinal perspective (i.e., ratio for a specific time-window) [21]. From a cross-sectional perspective, BoNT-A naive preschoolers (2–5 years old) with SCP had significantly lower MV growth rates in comparison to age-matched TD children (0.55 mL/month and 0.68 mL/month, respectively). From a longitudinal perspective over a time-window of 12 months, the TD children showed an MV growth rate of 0.90 mL/month while children with SCP presented an MV growth rate of 0.33 mL/month during the 12 months after BoNT-A injection [21].

Due to the pennate architecture of the MG muscle, MV growth is determined by increases in aCSA and muscle belly length (ML) [22]. Using a validated and reliable 3D freehand ultrasound (3DfUS) protocol, an ML increase of 6.9% was observed two weeks post-treatment. Moreover, increased fascicle lengths were reported by follow-ups ranging from one month to 12 months after the first BoNT-A treatment [21,23,24,25]. Furthermore, compared to TD children, higher echo-intensity (EI) values were observed in a cohort of children with a history of BoNT-A injections, and these were also significantly higher compared to BoNT-A naive children [26]. The latter suggested further decreased muscle integrity after recurrent BoNT-A treatment [26,27,28]. Additionally, this cross-sectional study observed larger MV deficits in children with a history of BoNT-A injections compared to a cohort of BoNT-A naive children, taking data of TD children as a reference [26].

Considering the evidence of early-altered muscle morphology, ambulatory children with SCP already bear an unfavorable muscle condition prior to initiating BoNT-A treatment [1,2]. The balance between the risk of BoNT-A worsening the impaired muscle growth and the functional benefits triggered by the reduced spasticity, remains unclear [13]. Previous muscle research warns for both the largest impact of the first BoNT-A exposure and the prolonged morphological alterations after recurrent BoNT-A injections. This phenomenon may consequently affect muscle strength and long-term functionality [18,26,29,30,31,32,33]. Therefore, the medium (i.e., six months) and/or long-term (i.e., one year) influence of both the initiation and recurrence of BoNT-A treatment on muscle morphology should be comprehensively assessed to understand the contribution of the neuro-paralytic working mechanism and post-injection recovery on hampered muscle growth. The lack of completely understanding the impact of BoNT-A treatment on MG muscle growth can be partially attributed to the study of incomplete datasets, such as only MV without reporting its determining variables (i.e., aCSA and ML). In addition, longitudinal data of a proper control group, i.e., children with SCP with no prescribed BoNT-A injection over time, is currently lacking. While no research could clearly delineate the actual course of muscle growth in children with SCP, BoNT-A injections hold a prominent role in the early treatment management of motor impairments. Therefore, improved insights into altered muscle growth over time starting from the first injection in the younger age groups and beyond the time-window of the neuro-paralysis due to the BoNT-A injection, are urgently needed to improve further treatment prescription and planning.

Therefore, the overall aim of this study was to investigate the effect of BoNT-A injection on morphological muscle growth in preschoolers and young children with SCP. Using 3DfUS, MG muscle morphology was assessed prior to the injection and six months after the injection. Morphological muscle outcomes were described in different dimensions including MV, ML, muscle–tendon unit complex length (MTUL), aCSA at 50% of the ML and EI (calculated within the defined aCSA). Based on the studies of Handsfield et al., muscle variables were normalized accounting for skeletal growth (e.g., aCSA normalized to body mass (aCSA_norm_) [34,35]. Following the approach by Barber et al., the baseline growth rate was expressed as the ratio of MV per age (cross-sectional perspective) and the growth rate over the six months follow-up time was expressed as the ratio of change in MV to the time of follow-up (longitudinal perspective) [21]. To achieve the overall aim, two research aims were established (Figure 1).

First, morphological changes between two time points (i.e., baseline and six months post-BoNT-A) for a group of BoNT-A naive children receiving their first BoNT-A session (i.e., *BoNT-A naive treatment group*), were compared with the morphological changes observed in an independent group of BoNT-A naive children with SCP who remained BoNT-A naive during the follow-up period (i.e., *BoNT-A naive untreated group*) (Research aim 1, Figure 1). The *BoNT-A naive untreated group* received usual care (i.e., physiotherapy and the use of orthoses) but had no clinical indication to prescribe BoNT-A treatment over a six-month period and were selected from ongoing muscle growth studies. This controlled, non-randomized research design incorporated unique data of the natural muscle growth in the SCP population within a timeframe of six months. The hypothesis stated that the initiation of BoNT-A treatment hampers MG muscle growth which was expected to be reflected by (a) lower amount of change in muscle morphology during the six month follow-up and (b) lower MV growth rates (expressed as ratio of change in MV to the time of follow-up) in the *BoNT**-A naive treatment* versus the *BoNT-A naive untreated group*. Second, the morphological changes for the *BoNT-A naive treatment group* were compared with morphological changes (i.e., between baseline and six months post-BoNT-A) in a group of children with SCP receiving a recurrent BoNT-A injection after a history of BoNT-A treatment (i.e., *BoNT-A not-naive treatment group*) (Research aim 2, Figure 1). This data analysis was performed only on normalized muscle morphology, to account for the impact of expected age differences between both groups. It was hypothesized that the initiation of BoNT-A treatment result in more pronounced reduction in MG muscle growth compared to recurrent BoNT-A treatment which was expected to be reflected by (a) lower amount of change in muscle morphology during the six months follow-up and (b) lower MV growth rates (expressed as ratio of change in MV to the time of follow-up) in the *BoNT-A naive treatment* versus the *BoNT-A not-naive treatment group*.

## 2. Results

### 2.1. Characteristics of the BoNT-A Treatment Groups and the Untreated Group

Ambulant children with SCP between two and nine years old and clinically planned for BoNT-A treatment were prospectively enrolled in two groups. First, eleven children were BoNT-A naive prior to the study inclusion (*BoNT-A naive treatment group*). An age- and severity-matched group of 11 children with SCP (median age of 2.8 y (IQR 1.8 y, gross motor function classification system (GMFCS)) I = 6, II = 1, III = 4) with neither a history of BoNT-A treatment nor a clinically prescribed BoNT-A treatment over 6-months’ time interval, was added as control (i.e., *BoNT-A naive untreated group*) [36]. Second, a group of 15 children with the clinically planned BoNT-A treatment had a history of BoNT-A injections (*BoNT-A not-naive treatment group*). The patient characteristics for all included children with SCP are presented in Table 1.

At the time of the baseline assessment, the usual care (e.g., use of orthoses and physiotherapy), functional mobility and clinical assessment of the ankle joint mobility and spasticity, were reviewed in the medical records (Appendix A). Regarding the treatment procedure for all children of the BoNT-A treatment groups, a median total dose of 2.8 units per kg body mass (IQR 1.0, minimum (min.) 0.9 to maximum (max.) 4.1 U/kg) BoNT-A (Botox^®^, Allergan, Diegem, Belgium) was injected in the MG muscle of the most involved leg (*n* = 26). Other injected muscles in the same leg were the psoas (*n* = 8), rectus femoris (*n* = 3) semitendinosus (*n* = 23), semimembranosus (*n* = 18), gracilis (*n* = 17), adductors (*n* = 3) and soleus muscle (*n* = 8) (Appendix A). No adverse events following the BoNT-A injection were reported. As part of the established integrated BoNT-A treatment approach at the CP Reference Centre of the University Hospitals Leuven, an intensive after-care program was prescribed including serial casting for min. 10 to max. 21 days (*n* = 23), the use of ankle–foot orthoses (*n* = 21) and regular physiotherapy (*n* = 26) [9]. The participants of the untreated group received usual care during the follow-up period (i.e., physiotherapy and the use of orthoses) (Appendix A).

Altered normalized MG muscle morphology in the *BoNT-A treatment groups* and *untreated group* were confirmed at the time of the baseline assessment with significant deficits in nMV, aCSA_norm_, normalized muscle lengths, EI and growth rates compared to TD children (Appendix A).

### 2.2. Comparison of BoNT-A Naive Treatment Group vs. Untreated Group (Research Aim 1)

At the start of the follow-up period, the children in the *BoNT-A*
*naive treatment group* and *untreated* SCP *group* did not differ in anthropometric and muscle outcomes (section 1 in Table 2). Changes in anthropometric and muscle outcomes during follow-up for both SCP groups are presented in section 2 in Table 2. The absolute and normalized muscle data at the time of the baseline as well as at the six months follow-up assessment for both SCP groups are presented in Table 3. During follow-up, the children with SCP increased significantly in body mass, body length and leg length compared to baseline (*p* = 0.003). This anthropometric growth was not significantly different between the *BoNT-A naive treatment* and *untreated* group (section 2 in Table 2).

Six months after BoNT-A treatment, the absolute MV did not change (+1%, *p* = 0.375), the nMV tended to decrease with 16% (*p* = 0.041) and the aCSA_norm_ significantly decreased with 17% (*p* = 0.003) in the BoNT-A naive treatment group (Table 3). For the latter, the amount of changes in muscle volume tended to be lower compared to morphological changes observed in the *BoNT-A naive untreated group* (MV: +1% vs. +13%, *p* = 0.045 and, nMV: −16% vs. −3%, *p* = 0.020, respectively) (Table 2 and Figure 2).

Furthermore, the changes in absolute and normalized aCSA were significantly lower while the change in EI was significantly higher in the *BoNT-A naive treatment group* compared to the children who remained BoNT-A naive (−6% vs. +12%, *p* = 0.003; −17% vs. +4%, *p* < 0.001 and +5% vs. −3% *p* = 0.009, respectively). The changes after six months in absolute and normalized muscle length outcomes did not significantly differ between the groups. The median MV growth rate during follow-up, calculated as the ratio of the change in MV to the six months of time-window, was 0.038 mL per month (IQR 0.85) in the *BoNT-A naive treatment group* which tended to be lower compared to MV growth rate during follow-up of the *BoNT-A naive untreated group* (median rate of 0.498 mL/month, IQR 0.46) (*p* = 0.026) (Figure 3).

### 2.3. Comparison of BoNT-A Naive Treatment Group vs. Not-Naive Treatment Group (Research Aim 2)

The morphological changes of the baseline and six months post-BoNT-A assessments were compared between the *BoNT-A naive treatment group* and the *BoNT-A not-naive treatment group.* The anthropometric and normalized muscle outcomes at the time of the baseline assessment and the changes over the six months post-BoNT-A follow-up period are presented in Table 4. Prior to injection (section 1 in Table 4), the *BoNT-A naive treatment group* was significantly younger and therefore characterized by significantly smaller anthropometrics measures in comparison to *the BoNT-A not-naive treatment* group (*p* < 0.001). Consequently, further comparisons were performed using only normalized muscle data. At the time of the baseline assessment, a tendency of higher EI values was observed in the children with a history of BoNT-A treatment compared to children with no earlier BoNT-A treatment (*p* = 0.047) (section 1 in Table 4). Six months after the current BoNT-A injection, no significant between-group differences were found for the amount of morphological muscle changes as well as MV growth rate during follow-up (section 2 in Table 4 and Figure 4).

## 3. Discussion

This prospective six-month follow-up study investigated the effect of BoNT-A injections on morphological muscle growth in BoNT-A naive preschool and young children with SCP. We aimed to compare (1) a group of preschool SCP children receiving their first injection (i.e., *BoNT-A naive treatment group*) and a matched control group of children with SCP that represent longitudinal natural course data without BoNT-A history (i.e., *BoNT-A naive untreated group*) and (2) the *BoNT-A naive treatment group* and a group of SCP children who were not BoNT-A naive receiving a recurrent BoNT-A injection (i.e., *BoNT-A not-naive treatment group*). We hypothesized that initiation of BoNT-A treatment hampers the MG muscle growth presented with (a) lower amount of change in muscle morphology during the six months follow-up and (b) lower MV growth rates. Already prior to the BoNT-A treatment, the children with SCP presented altered MG muscle growth which is in line with previous findings. The current results characterized the included study populations and further generalizing the observation of altered muscle morphology in CP.

To tackle the first research aim, no randomized control design could be implemented due to ethical reasons. However, the baseline muscle characteristics of both BoNT-A naive groups were comparable (*p* > 0.025, Table 2), confirming successful group matching. Nevertheless, in-depth analysis of clinical examination indicated more plantar flexor muscle spasticity in the *BoNT-A naive treatment group* compared to *the BoNT-A naive untreated group* (Appendix A). This was expected since BoNT-A treatment was clinically prescribed for these children. For the second research aim, the BoNT-A not-naive children with SCP received a clinically prescribed BoNT-A injection prior to the study inclusion, similar to the children of *the BoNT-A naive treatment group*. Due to the natural course in the presence of lower-limb spasticity, children with SCP are already treated with BoNT-A injection from the age of 2 to 4 years, and repeated BoNT-A injections are particularly prescribed for more involved children [7,37]. Consequently, the inclusion of children with a history of BoNT-A injection resulted in older ages at the time of study enrollment (age ranges of 3.8 to 9.1 years) with less motor abilities (GMFCS levels II and III for 67% of the children) in comparison to the children in the *BoNT-A naive treatment group* (age ranges from 2.8 to 4.9 years and 45% of the children with GMFCS level II and III).

### 3.1. Hampered Growth in Muscle Volume Six Months after BoNT-A Injection

The *BoNT-A naive treatment group* presented a more pronounced decrease in nMV (i.e., 16% less than the median baseline values) compared to the mean decrease of 10% reported in the study of Alexander et al., which also included only BoNT-A naive children [18]. Although it was hypothesized that BoNT-A injection would result in a lower amount of changes in volumes, the current 16% decrease in nMV was not significantly lower than the nMV muscle growth of untreated children over six-months (i.e., 3% decrease in MV). Interestingly, the six months effect of the BoNT-A injections presented no significant changes in absolute MV (1% increase, *p* = 0.375) suggesting that there was no muscle atrophy six months after the first BoNT-A injection. Within the group of untreated children with SCP, who represent the natural course of MV growth, a significant increase of 13% in MV over six months (*p* = 0.006) was found. This clear MV increase tended to be different from the MV changes of children in the *BoNT-A naive treatment* group (+13% vs. +1%, *p* = 0.045). These findings indicate that the initiation of BoNT-A treatment has hampered the growth in muscle volume. Yet, based on previous studies with cross-sectional designs, larger muscle alterations and neurogenic atrophy were found in children with high recurrence of BoNT-A injection [26,31]. Moreover, previous short-term follow-up studies (i.e., 13 weeks post-injection) also showed greater atrophy after the first exposure to BoNT-A [18,30]. In contrast, the initiation of BoNT-A treatment did not result in significantly more reduced MG muscle growth compared to recurrent BoNT-A treatment. It should be noted that the current study did not compare the effect of a new BoNT-A injection on muscle growth for children who had a history of BoNT-A injections to the muscle growth of matched children with BoNT-A history who are not planned for a new BoNT-A treatment. It was found to be too challenging to properly match such study groups, especially with respect to the number of previous BoNT-A sessions. Hence, future longitudinal studies should further delineate the potential specific impact of the first and second BoNT-A session on muscle growth within the same individuals.

To further understand the hampered muscle growth (expressed by altered MV), the changes in the longitudinal and cross-sectional muscle dimension should be explored. Indeed, the MG muscle architecture is defined by muscle fascicles and fibers of which the growth is represented by the number of sarcomeres in series and in parallel [38,39,40]. Methodologically, muscle fascicle lengths are challenging to measure in a reliable way using US, and were therefore not included in the current clinical follow-up study [41,42]. Alternatively, longitudinal muscle growth was approximated by muscle belly lengths. Similarly, the aCSA was used as an outcome parameter in the current study to estimate the muscle fiber growth in its cross-sectional dimension, as a surrogate to studying individual sarcomeres in parallel [24].

### 3.2. Preserved Growth in Muscle Lengths Six Months after BoNT-A Injection

The absolute ML and MTUL of the *BoNT-A naive treatment group* increased after six months (8%, *p* = 0.013 and 7%, *p* = 0.005, respectively) which was not significantly different from the untreated control group of SCP participants (*p* = 0.870 and *p* = 0.670), suggesting that muscle growth in longitudinal dimension was preserved. Regarding the normalized muscle lengths, median changes close to zero were presented in either the treated and untreated BoNT-A naive SCP group which suggests an equal longitudinal growth in muscle and bone lengths over six months. Reported inter-session intra-rater reliability data for nML and nMTUL in children with SCP presented standard errors of measurement (SEM) of 2% and 1%, respectively, which strengthens the conclusion that the longitudinal muscle growth was not hampered by BoNT-A injection [43]. To the best of our knowledge, no longitudinal data on muscle length changes following the first BoNT-A treatment have been reported to be suitable in comparison to the current results.

### 3.3. Reduced Anatomical Cross-Sectional Area Six Months after BoNT-A Injection

Within the *BoNT-A naive treatment group*, a reduction in aCSA (6%) and aCSA_norm_ (17%) was observed after the six months follow-up. These morphological muscle changes were significantly different from the increased cross-sectional growth of children who remained BoNT-A naive (aCSA, +12% and aCSA_norm,_ +4%) which highlighted a reduced cross-sectional growth six months post-BoNT-A injection (Table 2)**.** Taking into account the inter-session intra-rater SEM of 5%, these treatment-induced changes were considered meaningful [43]. Yet, only cross-sectional data were reported on aCSA deficits in children with SCP compared to TD children which limited the comparison of the current study results with previous findings only to the baseline data. Prior to injection, aCSA_norm_ deficits of 23% in the current *BoNT-A naive treatment group* versus the *TD group* were lower compared to the aCSA deficits of 28.5% for a previously studied cohort of SCP children with GMFCS levels II versus TD peers [44]. Pitcher et al. also reported aCSA deficits for SCP children with GMFCS level I compared to TD children, showing a deficit of 5% in aCSA for these 2–9 years old children with SCP. Notably, the current study group included children with GMFCS levels I, II and III. Appendix A presents the changes in aCSA_norm_ per GMFCS level and suggests the absence of differences between the GMFCS levels. However, larger sample sizes per GMFCS level are needed to further investigate the influence of the functional status on hampered muscle growth after BoNT-A injection.

### 3.4. Increased Echo-Intensity Values Six Months after BoNT-A Injection

The EI value (extracted from the defined aCSA at 50% of the mL) was studied as an indirect estimate of the altered structural integrity within the injected MG muscle [45,46]. Prior to the intervention, EI values were only significantly higher for children with a history of BoNT-A treatment compared to TD data (Appendix A). In contrast, these results were not observed for the baseline EI values of both (smaller) groups with BoNT-A naive children with SCP, suggesting that the muscle integrity was especially affected by recurrent BoNT-A injection sessions. These findings are in line with the reported positive association between BoNT-A history and echo-intensity observed in a previous cross-sectional investigation [26]. Six months after the BoNT-A injection, a trend of increased EI was identified in the BoNT-A naive treatment group. This change was only significantly different from the changes observed in the BoNT-A naive untreated group, and not from changes in the BoNT-A not-naive treatment group, highlighting the potential impact of the first BoNT-A treatment on structural muscle integrity. It should be noted that EI values are indirect general estimates of structural muscle integrity which cannot decompose underlying intramuscular changes such as the loss of contractile elements by replacement of fat and fibrotic tissue [13]. To directly determine the structural integrity within the injected muscle, animal and adult human studies employed ex vivo muscle biopsy analysis. The follow-up studies of 1–12 months post-BoNT-A injections indicated altered microscopic properties in animal muscles such as changes in muscle fiber CSA and fiber type distribution, damage of the fibrillar structures, increased amount of collagen and altered RNA profiles [15,16,17,40,47,48]. Moreover, a single study in SCP muscles showed a shift toward a faster profile in the medial gastrocnemius after recurrent BoNT-A injections [31]. Yet, the mechanisms involved in these intramuscular changes are not entirely understood. More research on microscopic muscle features in children with SCP are urgently needed to reveal the underlying mechanisms of mid- and long-term altered muscle growth in children with SCP after BoNT-A injection.

### 3.5. Muscle Growth Rates following BoNT-A Injection

For a more detailed understanding of the altered muscle growth, the baseline growth rate (ratio per baseline age) and growth rate during follow-up (ratio per specific follow-up period) were defined. Notably, this outcome assumes a constant linear growth. The calculation of muscle growth rate was introduced by Barber et al., who studied only BoNT-A naive children with SCP between the age of two to five years and presented a baseline growth rate of 0.55 mL/month [21]. In contrast, the current study groups of the same age range presented a lower median baseline growth rate in the three groups (0.47 mL/month (IQR 0.25), 0.51 mL/month (IQR 0.09), 0.42 mL/month (IQR 0.21) for the *BoNT-A naive treatment group*, the *BoNT-A naive untreated group* and *BoNT-A not-naive treatment group*, respectively). Although insignificant, lower MV growth rates during the six month follow-up period in the *BoNT-A naive treatment* vs. the *untreated group* were observed (0.04 mL/month (IQR 0.85) vs. 0.50 mL/month (IQR 0.46), respectively, *p* = 0.026). These results suggest that there is negligible growth in muscle volume following the first BoNT-injection, while the *BoNT-A naive untreated group* showed the expected muscle growth. Moreover, no differences in growth rates during follow-up were found between the *BoNT-A naive treatment* and n*ot-naive treatment* group (0.04 mL/month (IQR 0.85) vs. 2.10 mL/month (IQR 1.07), respectively, *p* = 0.838). It should be noted that the small sample sizes of all SCP groups combined with correction for multiple testing reduced the power of this analysis and may explain the lack of significant differences between the groups. Nevertheless, the pathophysiology of altered muscle growth remains unclear and is suggested to be defined by altered muscle use patterns and a number of factors including neuronal, nutritional and hormonal factors [22]. Further research is needed to reveal the consequence of either the presence of spasticity or the treatment of spasticity with BoNT-A injection on muscle growth.

### 3.6. Clinical Implications

The current findings suggested that the injected MG muscles of the current SCP cohort were not recovered after six months. Whereas the current *BoNT-A naive treatment group* showed a growth rate of 0.04 mL/month during the six months follow-up post-BoNT-A injections, a previous follow-up study spanning 12 months post-BoNT-A injection reported a longitudinal muscle growth rate of 0.33 mL/month [21]. These data suggest that muscle recovery continues at six months post-BoNT-A injection. In future studies, the muscle recovery profile should be further investigated over a longer time span (i.e., 12 months and longer). It is important to consider the post BoNT-A injection recovery phases for the planning and prescription of following BoNT-A treatments, especially in very young children who receive their first BoNT-A injection. The actual guidelines propose the application of an integrated multi-level BoNT-A treatment approach with conservative re-injection intervals of at least six months [8,9,49]. However, the current study results advise the postponement of re-injections beyond these six months. Close follow-up with morphological measurements of all injected muscles prior to and at regular intervals after BoNT-A injection is recommended. This might support the clinical decision-making process, and may eventually lead to longer intervals between re-injections [13,48]. It should be noted that research protocols have commonly set study inclusion criteria to the minimal time interval of six months after the previous BoNT-A injection [18,21,30]. The current findings suggest that, for future research, it might be reasonable to extend this interval as criterion for study inclusion.

As frequently cited, the muscle force generating capacity is related to the cross-sectional muscle dimensions highlighting the functional consequence of hampered cross-sectional growth [38,50]. Beyond the acute muscle atrophy enacted by the neuro-paralytic working mechanism of BoNT-A, the current study suggests that the impaired stimuli for muscle growth are likely to last longer than the clinical effects. However, in the current study, we analyzed only a limited set of clinical output parameters such as mobility and spasticity at the ankle joint prior to the injection. To further elaborate the mid/long-term benefits of BoNT-A treatment weighed against the harmful effects, comprehensive assessments of the motor outcomes (e.g., strength) and functional improvements (e.g., gait parameters) are advised to be included in future studies. The observation that growth in muscle lengths was not hampered by BoNT-A injection is promising, and strengthens the paradigm that BoNT-A treatment can avoid or delay muscle–tendon lengthening surgery [51]. However, the complexity in the pathophysiology of muscle contractures requires more accurate and dynamic assessments of the muscle lengths to support these suggestions.

## 4. Conclusions

Consistent with earlier evidence, the children with SCP and especially children for which BoNT-A treatment was clinically prescribed, displayed a predisposed altered muscle condition. Although absolute muscle atrophy was not determined, the current study revealed hampered medial gastrocnemius muscle growth six months post-BoNT-A injection. The hampered muscle growth was primarily attributed to the cross-sectional muscle dimension, while growth in the longitudinal muscle dimension remained preserved. In addition, the echo-intensity increased following BoNT-A injections (Figure 5). For the first time, a comparison to both (a) an untreated SCP cohort and (b) SCP cohort with a history of BoNT-A, using six months of follow-up data was investigated, suggesting BoNT-A injections further inhibit muscle growth, beyond the natural course of the pathology.

## 5. Materials and Methods

### 5.1. Participants

Patient recruitment was performed as part of an ongoing research project (Macroscopic and microscopic Muscle Architecture in cerebral Palsy, 3D-MMAP) between March 2019 and February 2021, with data collection completed in August 2021. The study protocol was approved by the Ethical Committee of the University Hospitals Leuven (three overlapping studies on muscle growth: S62187, S62645 and S59945). Written informed consent was obtained from the parents of all participants.

First, children with SCP who were clinically planned for BoNT-A treatment in the MG muscle were recruited for this study. In addition to the planned treatment, the eligibility for study inclusion was based on the following criteria: (a) diagnosis of spastic CP confirmed by a neuro-pediatrician, (b) aged between two and nine years, (c) GMFCS level I, II or III (d) no previous BoNT-A injection within the last six months, and (e) no history of orthopedic or neurological surgery. This resulted in the *BoNT-A*
*naive treatment* (i.e., no prior history of BoNT-A injection) and BoNT-A *not-naive treatment* (i.e., a history of previous BoNT-A injections) groups. Second, the research project database (S62187) of an ongoing longitudinal muscle growth study was consulted to select BoNT-A naive children who were not planned for BoNT-A treatment and who were group-matched based on the baseline age and the distribution in GMFCS levels of the children in the *BoNT-A naive treatment group* which resulted in a *BoNT-A naive untreated group*. Random selection was performed when more children were eligible for the group-matching. TD reference data of children with no history of neuromotor disorders were extracted from the established TD database related to running research projects (S62187, S62645 and S59945). Only the TD data with the following criteria were selected: (I) acquired with the same ultrasound settings as the SCP data, (II) data of children aged between 2 and 9 years in accordance with the age distribution of all children in the BoNT-A treatment groups.

### 5.2. Procedure

The muscles selected for injections and the total dose per muscle were empirically determined by a pediatric orthopedic specialist, based on the results of clinical examination and 3D gait analysis that are routinely planned prior to each BoNT-A treatment. The BoNT-A (Botox^®^, Allergan) injection was administered under general anesthesia and by guidance of ultrasound in a day-clinic setting.

### 5.3. Assessment

The participant’s anthropometry including body mass, body length and the length of the most involved leg were registered at baseline and at the follow-up. The leg length was measured from the lower border of the anterior superior iliac spine to the lower border of the medial malleolus. To further characterize the recruited patient groups clinically, results of the most recent clinical assessment for the ankle joint performed prior to the BoNT-A injection were extracted from the clinical patient medical record. The latter was also used to report the orthotic management, amount of physiotherapy and the use of medication. The median interval between the clinical assessment and the day of the BoNT-A injection was 49 days (IQR 55, min. 0 to max. 151 days). With the knee extended, maximal ankle dorsiflexion range of motion (ROM) was measured with a goniometer and the Modified Ashworth Scale (MAS) combined with the Tardieu R1 angle assessed the plantar flexor spasticity [52,53,54]. Furthermore, the (a) ankle joint mobility and (b) plantar flexor spasticity were classified as follows (a) no contracture (≥10 degrees of dorsiflexion), mild restriction in ROM (10 to 0 degrees of dorsiflexion), contracture (<0 degrees of dorsiflexion) and (b) normal to mild increased tone (MAS of 0 or 1), moderate spasticity (MAS of 1+ or 2) and severe spasticity (MAS of 3 or more). These clinical results were also used to define the most involved side for all SCP participants. In addition to the GMFCS scale, the Functional Mobility Scale scored the walking ability over three distances (i.e., 5–50–500m) on a six-point scale for which lower scores indicate a greater need for assistance [55]. For the TD children, the measured leg was randomly selected.

A validated and reliable 3D freehand ultrasound protocol was applied to assess the MG muscle morphology [26]. Before the acquisition was performed, the participants were placed in a prone position with a small triangular cushion under the lower leg to allow knee flexion and a resting position of the ankle over the edge of the cushion. The angles of both joints were measured with a goniometer and resulted in a median angle of 25 degrees (IQR 5) knee flexion and 35 degrees (IQR 10) ankle plantarflexion. STRADWIN software (version 6.0; Mechanical Engineering, Cambridge University, Cambridge, UK) was used during both data acquisition and processing. The ultrasound acquisition, which started at the medial femoral condyle and continued to the distal end of the calcaneus, was performed by experienced examiners. Large amounts of acoustic transmission gel were used and combined with the Portico (i.e., custom-shaped plastic mount combined with gel pad) which minimizes muscle deformation during ultrasound acquisitions [56]. The acquired images of the automatically generated 3D muscle reconstruction were manually analyzed by a single experienced researcher [57]. Prior to the statistical analysis, this processor was blinded for the patient sensitive data, the timing of assessment and the group allocation. Manual segmentations along the inner muscle borders were drawn at 5–10% of the cross-sectional images in order to calculate the muscle volume (milliliter, mL) by the cubic planimetry technique [58]. The MV was used to calculate the baseline growth rate expressed as a ratio of MV (mL) per age (month). The MV growth rate during the follow-up period was calculated as the ratio of the change in MV (mL) to the time of follow-up interval (months). To extract the muscle lengths (in mm), three anatomical landmarks were visually identified. This resulted in the muscle origin at the most superficial aspect of the medial formal condyle (landmark A); the muscle tendon junction (MTJ) which is the transition of the muscle belly insertion and the tendon origin (landmark B) and the tendon insertion at the first proximal image of the calcaneus (landmark C). Muscle belly length (ML) was defined as the distance between landmarks A and B, muscle tendon length (TL) was calculated between landmark B and C and the muscle–tendon unit complex length (MTUL) was calculated between landmarks A and C. At 50% of ML, the inner muscle border was specifically segmented on a transverse ultrasound image to extract the anatomical cross-sectional area (aCSA) of which the echo-intensity (EI) value was also exported [4]. In case of noisy muscle images and/or difficult visualization of the landmarks, a second trained processor was consulted. To account for anthropometric variation and skeletal growth over the study period, MV was normalized to body mass multiplied by length, based on the equation of Handsfield et al. (2016) [35]. Muscle lengths and aCSA were normalized to leg length and body mass, respectively.

### 5.4. Statistical Analysis

Normality of the outcomes were visually checked and statistically confirmed by the Shapiro–Wilk test (*p* > 0.05). Outliers were identified by inspection of a boxplot. Data are presented by median (interquartile ranges), unless otherwise stated. Significance level of 0.01 was used after Bonferroni correction for multiple comparisons, based on the following muscle outcomes: MV, ML, MTUL, aCSA and EI. All statistical tests were performed in the SPSS Software platform (Version 28.0, IBM, Armonk, NY, USA). Figures were designed in GraphPad Prism Software (Version 9, San Diego, CA, USA)

To compare the morphological changes over two time points (i.e., baseline and six months post-BoNT-A) between *the BoNT-A naive treatment* and (a) *BoNT-A naive untreated group* and (b) *BoNT-A not-naive treatment group*, a Mann–Whitney U test was employed. Within *the BoNT-A naive treatment group*, the differences in morphological muscle outcomes between the baseline and follow-up assessment were determined using Wilcoxon signed-rank test. Kruskal–Wallis test with pairwise post hoc analysis compared the altered muscle morphology between the three *SCP groups* and the TD group at the time of the baseline assessment.

## Figures and Tables

**Figure 1 toxins-14-00139-f001:**
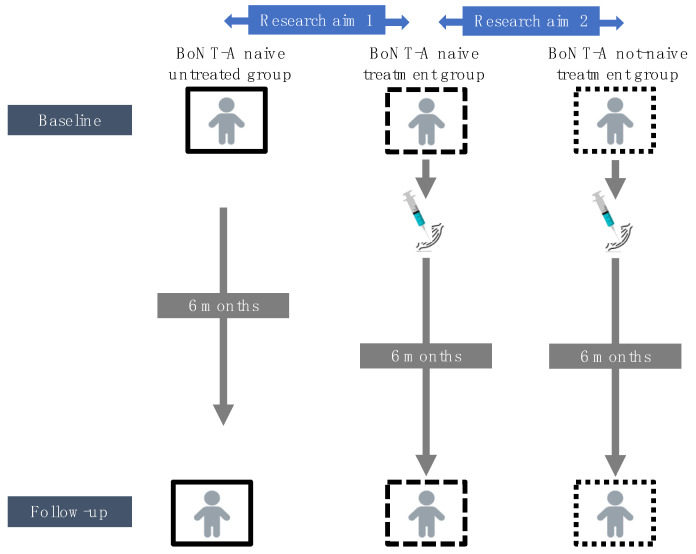
Schematic overview of the research aims. BoNT-A, botulinum neurotoxin type-A.

**Figure 2 toxins-14-00139-f002:**
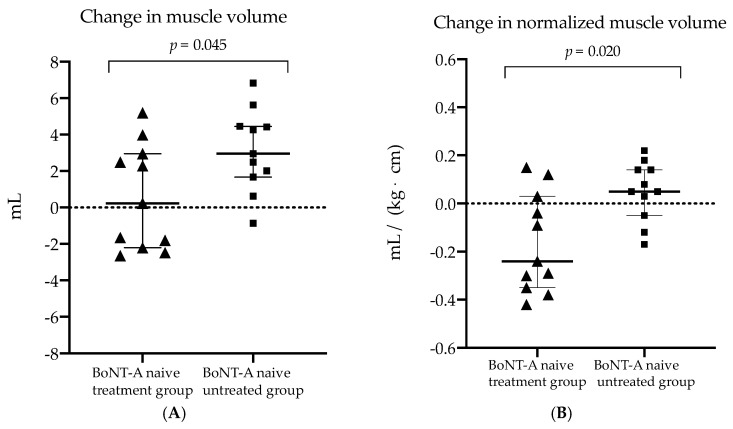
Scatter dot plot with median, interquartile range and individual values of (**A**) the change in muscle volume and (**B**) change in the normalized muscle volume (normalized to product for body mass and length) for the botulinum neurotoxin type-A (BoNT-A) naive treatment (triangls) and BoNT-A naive untreated group (squares). mL, milliliter; kg, kilogram; cm, centimeter.

**Figure 3 toxins-14-00139-f003:**
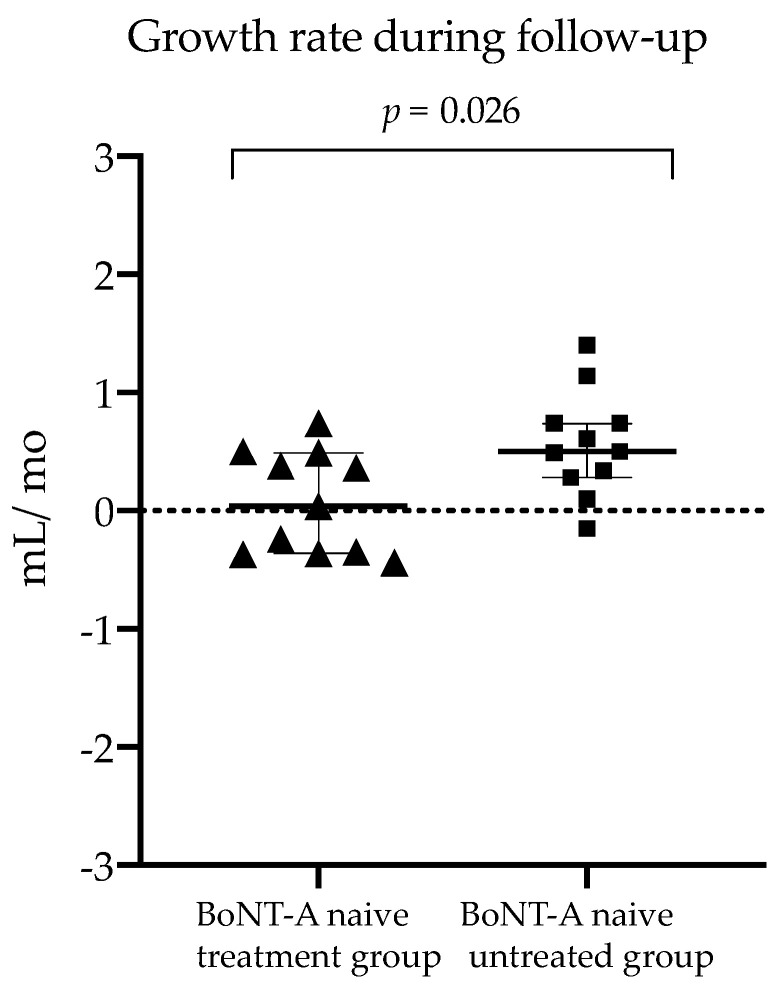
Scatter dot plot with median, interquartile range and individual values of the muscle volume growth rate during follow-up for the botulinum neurotoxin type-A (BoNT-A) naive treatment (triangles) and BoNT-A untreated group (squares). Muscle growth rate was calculated as ratio of change in muscle volume (milliliter) per time of follow-up (months). mL, milliliter; mo, months.

**Figure 4 toxins-14-00139-f004:**
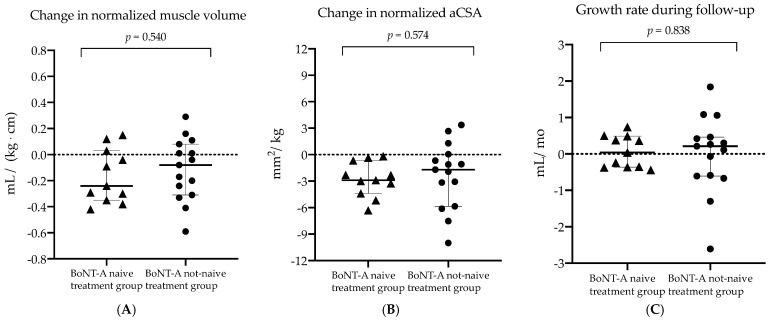
Scatter dot plot with median, interquartile range and individual values of (**A**) the change in normalized muscle volume (normalized to product of body mass and length), (**B**) change in the normalized anatomical cross-sectional area (normalized to body mass) and (**C**) growth rate during follow-up for the botulinum neurotoxin type-A (BoNT-A) naive treatment (triangles) and BoNT-A not-naive treatment group (dots). mL, milliliter; kg, kilogram; cm, centimeter; mm, millimeter; mo, months.

**Figure 5 toxins-14-00139-f005:**
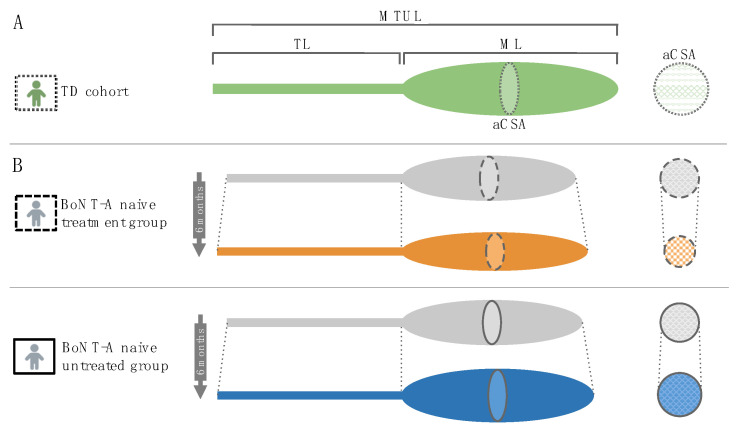
Schematic representation of the main findings in the current stud. On top (**A**), the medial gastrocnemius muscle of typically developing (TD) children is represented including the key morphological outcomes: muscle–tendon unit complex length (MTUL), tendon length (TL), muscle belly length (ML) and anatomical cross-sectional area (aCSA). The latter is represented separately using circle areas for clarity. The echo-intensity is represented with a pattern fill of the aCSA. Below (**B**), the comparison in outcomes prior to and six months after BoNT-A injection is shown for the botulinum neurotoxin type-A (BoNT-A) naive treatment group indicating the reduced aCSA and increased muscle lengths. At the bottom, the comparison in outcomes over six months follow-up for the BoNT-A naive children who remained treatment naive is shown, indicating increased cross-sectional and longitudinal muscle growth.

**Table 1 toxins-14-00139-t001:** Patient characteristics of the BoNT-A naive treatment (*n* = 11), BoNT-A naive untreated (*n* = 11) and BoNT-A not-naive treatment group (*n* = 15) at the time of the baseline assessment.

Group	BoNT-A Naive Treatment	BoNT-A Naive Untreated	BoNT-A Not-Naive Treatment
Age (years)	3.3 (1.2)	2.8 (1.8)	6.4 (2.5)
Topographic involvement	Bilateral, *n* = 7 Unilateral, *n* = 4	Bilateral, *n* = 4 Unilateral, *n* = 7	Bilateral, *n* = 8 Unilateral, *n* = 7
Gender	Boy, *n* = 8 Girl, *n* = 3	Boy, *n* = 4 Girl, *n* = 7	Boy, *n* = 8 Girl, *n* = 7
GMFCS level	I, *n* = 6 II, *n* = 1 III, *n* = 4	I, *n* = 6, II, *n* = 1 III, *n* = 4	I, *n* = 5 II, *n* = 8 III, *n* = 2
Previous BoNT-A sessions	NA	NA	1–2, *n* = 9 3–6, *n* = 6
Time between two assessments (weeks)	27.4 (3.4)/24–36	25 (2.5)/20–30	26.7 (3.6)/23–35

Age is presented as median (interquartile range). Time between assessments is presented as median (interquartile range)/minimum-maximum. BoNT-A, botulinum neurotoxin type-A; *n*, number, GMFCS, gross motor function classification system; NA, not applicable.

**Table 2 toxins-14-00139-t002:** Comparison of median group values for anthropometric measures and medial gastrocnemius muscle outcomes between the *BoNT-A naive treatment group* (*n* = 11) (a) and *BoNT-A naive untreated group* (*n* = 11) (b) for the baseline assessment (section 1) and the changes over the six-months follow-up (section 2).

	BoNT-A Naive Treatment (a)	BoNT-A Naive Untreated (b)	*p*
Section 1: Baseline assessment			
Age (years)	3.3 (1.2)	2.8 (1.8)	0.401
Body mass (kg)	13.6 (1.3)	14.5 (4.9)	0.243
Body length (cm)	91.0 (9.2)	97.0 (18.4)	0.652
Leg length (cm)	44.2 (5.0)	46.0 (9.5)	0.797
MV (mL)	16.6 (8.80)	22.8 (14.9)	0.300
nMV (mL/(kg·m))	1.47 (0.43)	1.52 (0.45)	0.478
ML (mm)	104.3 (19.0)	111.0 (33.5)	0.652
nML (%)	23.6 (4.05)	23.3 (5.83)	0.949
MTUL (mm)	203.1 (25.8)	207.8 (57.5)	0.748
nMTUL (%)	45.8 (1.92)	46.7 (5.55)	0.438
aCSA (mm^2^)	245.7 (79.4)	258.1 (94.3)	0.652
aCSA_norm_ (mm^2^/kg)	16.7 (6.40)	17.5 (4.15)	0.797
EI (bit)	153.2 (12.1)	158.5 (31.5)	1.000
Growth rate (mL/mo)	0.47 (0.25)	0.51 (0.09)	0.270
Section 2: Changes over six-month follow-up			
Interval (months)	6.00 (1.00)	6.00 (0.00)	0.075
Δ Body mass (kg)	1.20 (1.20)/+8%	1.10 (0.65)/+ 8%	0.217
Δ Body length (cm)	5.60 (3.90)/+6%	4.20 (3.20)/+ 4%	0.365
Δ Leg length (cm)	2.80 (2.80)/+6%	2.50 (2.20)/+5%	0.847
Δ MV (mL)	0.23 (5.15)/+1%	2.95 (2.78)/+13%	0.045
Δ nMV (mL/(kg⋅m))	−0.24 (0.38)/−16%	0.05 (0.19)/−3%	0.020
Δ ML (mm)	8.50 (9.10)/+8%	7.87 (9.00)/+7%	0.870
Δ nML (%)	0.47 (2.67)/+2%	0.78 (2.67)/+3%	0.818
Δ MTUL (mm)	13.1 (9.40)/+7%	8.90 (13.8)/+4%	0.670
Δ nMTUL (%)	−0.56 (3.63)/−1%	−0.82 (2.08)/−2%	0.622
Δ aCSA (mm^2^)	−14.2 (49.9)/−6%	30.0 (23.1)/+12%	0.003
Δ aCSA_norm_ (mm^2^/kg)	−2.89 (3.72)/−17%	0.72 (3.18)/+4%	<0.001
Δ EI (bit)	7.80 (23.7)/+5%	−5.30 (23.2)/−3%	0.009
Growth rate during follow-up (Δ mL/Δ mo)	0.038 (0.85)	0.498 (0.46)	0.026

Data are presented as median (interquartile range)/percentage. The percentual differences were calculated based on the median difference scores and expressed relative to the median baseline data. The +/− refers to positive/negative differences compared to baseline data. Muscle growth rate was calculated as ratio of muscle volume (milliliter) per age (months). The growth rate during the follow-up period was calculated as the ratio of the change in muscle volume (mL) to the time of follow-up (months). Mann-Whitney U test with significance level of 0.01 was used after Bonferroni correction. Significant results were indicated with grey color. BoNT-A, botulinum neurotoxin type-A; *n*, number; kg, kilograms; cm, centimeter; mL, milliliter; nMV, muscle volume normalized to the product of body mass and body length; ml, milliliter; m, meter; mm, millimeter; nML, muscle belly length normalized to leg length; nMTUL, muscle–tendon unit complex length normalized to leg length; aCSA_norm_, anatomical cross-sectional area normalized to body mass; EI, echo-intensity; mo, month.

**Table 3 toxins-14-00139-t003:** Comparison of median values between the two assessments for medial gastrocnemius outcomes in the *BoNT-A naive treatment group* compared to the *BoNT-A naive untreated group*.

	BoNT-A Naive Treatment (a)				BoNT-A Naive Untreated (b)			
	Baseline	Follow-Up	Median Differences	*p*	Baseline	Follow-Up	Median Differences	*p*
Participants (*n*)	11	11			15	15		
Age	3.3 (1.2)	3.8 (1.0)	0.60	0.003	2.8 (1.8)	3.4 (1.8)	0.60	0.003
Body mass (kg)	13.6 (1.3)	14.7 (2.6)	1.20 (+8%)	0.003	14.5 (4.9)	15.6 (4.3)	1.10 (+8%)	0.003
Body length (cm)	91.0 (9.2)	99.0 (9.0)	5.60 (+6%)	0.003	97.0 (18.4)	99.0 (14.9)	4.20 (+4%)	0.003
Leg length (cm)	44.2 (5.0)	46.5 (8.0)	2.80 (+6%)	0.003	46.0 (9.5)	49.0 (10.0)	2.50 (+5%)	0.003
MV (mL)	16.6 (8.80)	16.6 (13.4)	+0.23 (+1%)	0.375	22.8 (14.9)	27.1 (12.6)	+2.95 (+13%)	0.006
nMV (mL/(kg⋅m))	1.47 (0.43)	1.24 (0.43)	−0.24 (−16%)	0.041	1.52 (0.45)	1.49 (0.51)	−0.05 (−3%)	0.155
ML (mm)	104.3 (19.0)	107.5 (21.5)	+8.50 (+8%)	0.013	111.0 (33.5)	115.1 (29.4)	+7.87 (+7%)	0.007
nML (%)	23.6 (4.05)	23.5 (2.41)	+0.47 (+2%)	0.534	23.3 (5.83)	23.5 (5.23)	+0.78 (+3%)	0.790
MTUL (mm)	203.1 (25.8)	211.8 (35.2)	+13.1 (+7%)	0.005	207.8 (57.5)	221.2 (45.4)	+8.90 (+4%)	0.008
nMTUL (%)	45.8 (1.92)	45.2 (2.50)	−0.56 (−1%)	0.859	46.7 (5.55)	46.3 (6.04)	−0.82 (−2%)	0.374
aCSA (mm^2^)	245.7 (79.4)	224.9 (101.0)	−14.2 (−6%)	0.075	258.1 (94.3)	295.7 (142.1)	+30.0 (+12%)	0.013
aCSA_norm_(mm^2^/kg)	16.7 (6.40)	13.5 (7.98)	−2.89 (−17%)	0.003	17.5 (4.15)	18.8 (5.82)	+0.72 (+4%)	0.110
EI (bit)	153.2 (12.1)	162.6 (18.8)	+7.80 (+5%)	0.033	158.5 (31.5)	153.1 (17.9)	−5.30 (−3%)	0.131

Data are presented as median (interquartile range). Median differences were calculated based on the median of the individual scores per outcome and expressed as both absolute and percentual changes relative to the median baseline data. The +/− refers to positive/negative differences compared to baseline data. Wilcoxon signed-rank test with significance level of 0.01 was used after Bonferroni correction. Significant results were indicated with grey color. BoNT-A, botulinum neurotoxin type-A; *n*, number; kg, kilograms; cm, centimeter; mL, milliliter; nMV, muscle volume normalized to the product of body mass and body length; m, meter; mm, millimeter; nML, muscle belly length normalized to leg length; nMTUL, muscle–tendon unit complex length normalized to leg length; aCSA_norm_, anatomical cross-sectional area normalized to body mass; EI, echo-intensity.

**Table 4 toxins-14-00139-t004:** Comparison of median group values for anthropometric measures and medial gastrocnemius normalized muscle outcomes between the *BoNT-A naive treatment group* (*n* = 11) (a) and *BoNT-A not-naive treatment group* (*n* = 15) (b) for the baseline assessment (section 1) and the changes over the six-months follow-up (section 2).

	BoNT-A Naive Treatment (a)	BoNT-A Not-Naive Treatment (b)	*p*
Section 1: Baseline assessment			
Age (years)	3.3 (1.2)	6.4 (2.5)	<0.001
Body mass (kg)	13.6 (1.3)	20.0 (7.5)	<0.001
Body length (cm)	91.0 (9.20)	114.0 (26.0)	<0.001
Leg length (cm)	44.2 (5.0)	55.4 (15.5)	<0.001
nMV (mL/(kg⋅m))	1.47 (0.43)	1.39 (0.62)	0.959
nML (%)	23.6 (4.05)	23.4 (3.46)	0.838
nMTUL (%)	45.8 (1.92)	45.5 (1.49)	0.384
aCSA_norm_ (mm^2^/kg)	16.7 (6.40)	17.8 (5.56)	0.721
EI (bit)	153.2 (12.1)	175.3 (22.4)	0.020
Growth rate (mL/mo)	0.47 (0.25)	0.42 (0.21)	0.959
Section 2: Changes over six-month follow-up			
Interval (months)	6.00 (1.00)	7.00 (1.00)	
Δ Body mass (kg)	1.20 (1.20)/(+8%)	1.20 (1.00)/(+6%)	0.959
Δ Body length (cm)	5.60 (3.90)/(+6%)	3.20 (2.30)/(+3%)	0.097
Δ Leg length (cm)	2.80 (2.80)/(+6%)	3.00 (1.10)/(+5%)	1.000
Δ nMV (mL/(kg⋅m))	−0.24 (0.38)/−16%	−0.08 (0.39)/−6%	0.540
Δ nML (%)	0.47 (2.67)/+2%	−0.15 (0.82)/−1%	0.474
Δ nMTUL (%)	-0.56 (3.63)/−1%	0.02 (1.79)/0%	1.000
Δ aCSA_norm_ (mm^2^/kg)	−2.89 (3.72)/−17%	−1.69 (1.73)/−10%	0.574
Δ EI (bit)	7.80 (23.7)/+5%	9.20 (36.2)/+5%	1.000
Growth rate during follow-up (Δ mL/Δ mo)	0.038 (0.85)	0.210 (1.07)	0.838

Data are presented as median (interquartile range)/percentage. The percentual differences were calculated based on the median difference scores and expressed relative to the median baseline data. The +/− refers to positive/negative differences compared to baseline data. Muscle growth rate was calculated as ratio of muscle volume (milliliter) per age (months). The growth rate during the follow-up period was calculated as the ratio of the change in muscle volume (mL) to the time of follow-up (months). Mann-Whitney U test with significance level of 0.01 was used after Bonferroni correction. Significant results were indicated with grey color. BoNT-A, botulinum neurotoxin type-A; *n*, number; kg, kilograms; cm, centimeter; mL, milliliter; nMV, muscle volume normalized to the product of body mass and body length; mL, milliliter; m, meter; mm, millimeter; nML, muscle belly length normalized to leg length; nMTUL, muscle–tendon unit complex length normalized to leg length; aCSA_norm_, anatomical cross-sectional area normalized to body mass; EI, echo-intensity; mo, month.

## Data Availability

The raw data supporting the conclusions of this article will be made available by the authors, without undue reservation.

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
