# Peer review of "Reduced Cross-Sectional Muscle Growth Six Months after Botulinum Toxin Type-A Injection in Children with Spastic Cerebral Palsy"

_toxins, 2022, doi:10.3390/toxins14020139_

Round 1

Reviewer 1 Report

Excellent reorganization and presentation of the data

Reviewer 2 Report

Elegant and convincing, both  data and presentation.

Reviewer 3 Report

This study explores the consequences of BoNT-A injections on the medial gastrocnemius of a population of cerebral plays children, comparing a cohort BoNT-A naive treatment to BoNT-A naive untreated and BoNT-A non naive treatment over 6 months follow up. The findings with reduced cross sectional growth after injections and preservation of muscle length lead to the recommendation of keeping a 6 months interval between injections in CP children, in addition, interestingly the trend of increase echo- intensity values after the initial first injection was found compared to BoNT-A naive untreated but not to BoNT-A non naive treatment, pointing  towards the impact of the first injection on muscle integrity.

It’s a very interesting study, with good methodology and data analysis and appropriate conclusions.

This manuscript is a resubmission of an earlier submission. The following is a list of the peer review reports and author responses from that submission.

Round 1

Reviewer 1 Report

This manuscript is well written. The topic is of interest to this population of children. The question is whether the risks are worth the benefit. The discussion could be reduced in length. 

Reviewer 2 Report

BoNT is commonly used for spasticity management. It is important understand its effect of spastic muscles. This study aimed to tackle this clinically important problem. 

However, I have serious concerns regarding experimental design. 

1) Aim 1 compared TD and BoNT-A treatment group. the problem is that BoNT-A treatment group is a mix of  BoNT-Naive SCP patients and patients who received different number of previous injections. such comparisons did not give any useful information.

2) Aim 2 compared pre- and post-BoNT assessment in the BoNT-treatment group, again, which is a mix of naive and non-naive patients. Again, such comparisons did not provide any useful information.

3) Aim 3 compared TD and naive patients at baseline and 6 months after BoNT. naive-control patients (untreated) were also included. This aim is informative. the problem is that there are some statistically significant differences in baseline assessments between Naive-treatment group and naive-untreated group. 

Overall, aims 1 & 2 need to be deleted. Aim 3 is informative. 

Reviewer 3 Report

This manuscript presents a well thought-out and carefully designed study of children with SCP treated with botulinum toxin. The paper was a pleasure to read and presents their findings comprehensively, particularly in comparison to their control groups and other works already published.

My comment are minor.

The Title does not read well and is a little confusing to the reader. I recommend that this be adjusted.

Figure 1 could be clearer

Tables 1, S1, S2      The other data are presented as frequencies…not required in the legends as n is given

Lines 147-152         How were the doses distributed between the muscles?

Line 324                  Citation needed

The authors have described two findings that seem at odds with each other.  At lines 325-326, they report These findings indicate that the initiation of BoNT-A treatment has hampered the muscle growth.  At lines 343-346 they state Regarding the normalized muscle lengths, median changes close to zero were presented in either the treated and untreated SCP group which suggests an equal growth in muscle and bone lengths over six months.  At lines 347-348 they state ….strengthens the conclusion that the muscle length growth was not hampered by BoNT-A injection.  The authors should clarify.

Lines 363-364         the amount of decrease naCSA was enlarged to 17%....this should be rewritten

Lines 393-394         The authors should make clear that some of the citations used here are for animal studies.

Figure 4                   For clarity, the parts of this figure could be labelled a), b) and c)

Lines 457-457         The authors include a comment about 12 months, but their study is only for 6 months.

Reviewer 4 Report

It's an article that will surely interest the doctors that apply botulinum toxin injections.

Additionally, you can find my minor corrections in the PDF attached. 
